DATA RELEASE

# The genome assembly and annotation of the many-banded krait, *Bungarus multicinctus*

Boyang Liu[1,†], Liangyu Cui[1,†], Zhangwen Deng[2,†], Yue Ma[1], Diancheng Yang[3,4], Yanan Gong[3,4], Yanchun Xu[1], Tianming Lan[5,6], Shuhui Yang[1,*] and Song Huang[3,4,*]

1 College of Wildlife and Protected Area, Northeast Forestry University, Harbin 150040, China
2 Guangxi Forest Inventory and Planning Institute, Nanning 530011, China
3 Anhui Province Key Laboratory of the Conservation and Exploitation of Biological Resource, College of Life Sciences, Anhui Normal University, Wuhu 241000, China
4 Huangshan Noah Biodiversity Institute, Huangshan 245000, China
5 BGI Life Science Joint Research Center, Northeast Forestry University, Harbin 150040, China
6 State Key Laboratory of Agricultural Genomics, BGI-Shenzhen, Shenzhen 518083, China

## ABSTRACT

Snakes are a vital component of wildlife resources and are widely distributed across the globe. The many-banded krait *Bungarus multicinctus* is a highly venomous snake found across Southern Asia and central and southern China. Snakes are an ancient reptile group, and their genomes can provide important clues for understanding the evolutionary history of reptiles. Additionally, genomic resources play a crucial role in comprehending the evolution of all species. However, snake genomic resources are still scarce. Here, we present a highly contiguous genome of *B. multicinctus* with a size of 1.51 Gb. The genome contains a repeat content of 40.15%, with a total length exceeding 620 Mb. Additionally, we annotated a total of 24,869 functional genes. This research is of great significance for comprehending the evolution of *B. multicinctus* and provides genomic information on the genes involved in venom gland functions.

**Subjects** Genetics and Genomics, Evolutionary Biology, Zoology

**Submitted:** 10 February 2023

\* Corresponding authors. E-mail: snakeman@ahnu.edu.cn; yangshuhui@nefu.edu.cn

† Contributed equally.

Preprint submitted at https://doi.org/10.20944/preprints202306.0658.v1

Included in the series: *Snake Genomes* (https://doi.org/10.46471/GIGABYTE_SERIES_0003)

*Gigabyte*, 2023, 1–**??**

## INTRODUCTION

Snakes are a fascinating group of reptiles that exhibit unique and diverse characteristics. With approximately 3,070 extant species in all continents except Antarctica [1], they are known for lacking limbs, elongated body shapes, and an exclusively carnivorous diet. Snakes have evolved many specialized adaptations, such as infrared sensing pits and a venom apparatus, which provide them with exceptional predatory capabilities [1]. These adaptations have made snakes important model organisms for evolutionary studies, yielding insights into limb development, sex chromosome evolution, and venom evolution. In recent years, genetic approaches have become increasingly important in understanding the evolution and diversity of snakes [2]. By exploring the evolution of venomous snakes, we can gain a deeper understanding of the ecological and evolutionary roles of these intriguing species.

*Bungarus multicinctus* (NCBI:txid8616), also known as the many-banded krait or umbrella snake, is widely distributed throughout southern Asia, its range spanning across countries such as India, Pakistan, Indonesia, Sri Lanka, Malaysia, Bangladesh, Vietnam, and

**Table 1.**  Summary of the features of our *B. multicinctus* genome.

|  | Contig | Scaffold |
|---|---|---|
| Maximal length (bp) | 468,983 | 41,606,426 |
| N90 (bp) | 5,806 | 30,083 |
| N50 (bp) | 33,081 | 6,870,761 |
| Number ≥ 100bp | 163,090 | 82,383 |
| Number ≥ 2kb | 81,775 | 22,350 |
| Ratio of Ns | 0.045 | 0.045 |
| GC content (%) | 39.6 | 37.8 |
| Genome size (bp) | 1,548,488,562 | 1,621,955,402 |

**Table 2.**  Summary of TEs in our *B. multicinctus* genome.

| Type | Repbase TEs | | TE proteins | | *De novo* | | Combined TEs | |
|---|---|---|---|---|---|---|---|---|
|  | Length (bp) | % in genome | Length (bp) | % in genome | Length (bp) | % in genome | Length (bp) | % in genome |
| DNA | 32,816,331 | 2.02 | 2,921,569 | 0.18 | 112,067,211 | 6.91 | 129,267,220 | 7.97 |
| LINE | 174,481,405 | 10.76 | 154,961,354 | 9.60 | 276,722,230 | 17.07 | 301,624,987 | 18.61 |
| SINE | 13,524,698 | 0.83 | 0 | 0 | 39,754,823 | 2.45 | 43,837,124 | 2.70 |
| LTR | 23,313,679 | 1.44 | 30,431,704 | 1.88 | 52,496,522 | 3.24 | 60,898,786 | 3.76 |
| Other | 16,171 | 0.01 | 243 | 0.01 | 0 | 0 | 16,414 | 0.01 |
| Unknown | 0 | 0 | 0 | 0 | 182,574,604 | 11.26 | 182,574,604 | 11.26 |
| Total | 234,804,260 | 14.49 | 188,249,038 | 11.61 | 645,464,460 | 39.82 | 675,577,436 | 41.68 |

China [3]. *B. multicinctus* is recognized as one of the ten most venomous snakes in China, with a lethality rate ranging from 26.9% to 33.3% [4].

In this study, we collected a muscle sample of *B. multicinctus* to generate a highly contiguous genome with a genome size of 1.51 Gb. Its repeat element content reached 41.68%, providing new evidence for understanding the relationship between repeat elements and genome size in Elapidae species.

## MAIN CONTENT

### Context

This study presents a highly continuous genome assembly of *B. multicinctus*. The genome size of *B. multicinctus* was found to be 1.51 Gb, with a GC content of 37.8% (Table 1). The maximal scaffold length was 39.68 Mb, and the N50 length was 6.55 Mb, indicating a highly continuous genome sequence. This draft genome sequence of *B. multicinctus* will serve as an invaluable resource for further research on venomous snakes, enabling a better understanding of their genetic makeup.

The content of repetitive elements in our *B. multicinctus* genome was surprisingly large, reaching 41.68% with a total length of 675 Mb (Table 2). We analyzed the content of various repeating elements. While unknown types accounted for 51% of the repeating elements, LINEs and DNA transposons accounted for 10% and 8%, respectively (Figure 1). Research indicates that although snake species have similar genome sizes, they exhibit significant differences in TE content, with low diversity in the types of TEs [5]. Specifically, species with a longer evolutionary history tend to have higher TE diversity [6]. Our results suggest that the significant expansion of repeating elements is an important manifestation of species differences.

We identified 24,869 functional genes of *B. multicinctus* and annotated them with Kyoto Encyclopedia of Genes and Genomes (KEGG). The majority of these genes were found to be involved in pathways related to Environmental Information Processing and Metabolism.



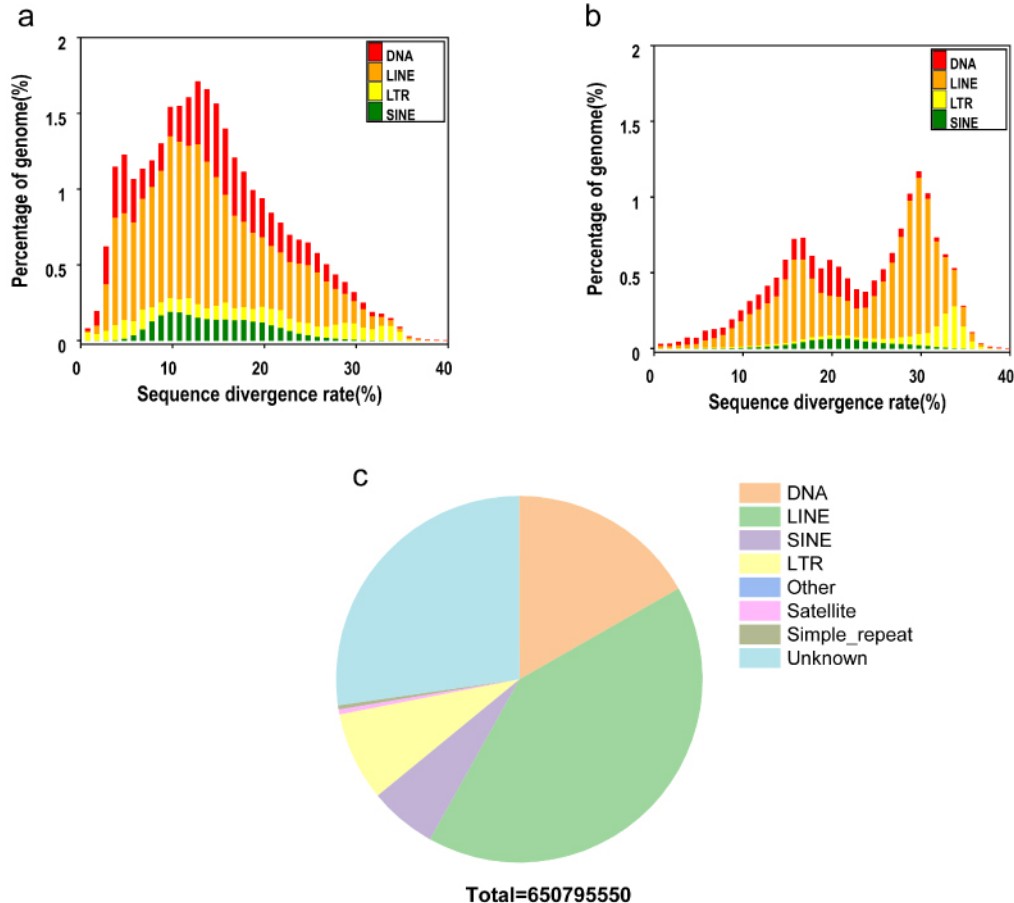

**Figure 1.** Distribution of transposable elements (TEs) in the *B. multicinctus* genome. The TEs include DNA transposons (DNA) and RNA transposons (i.e., DNAs, LINEs (Long Interspersed Nuclear Elements), LTRs (Long Terminal Repeats), and SINEs (Short interspersed nuclear elements)). (a) Distribution of the *de novo* sequence divergence rate. (b) Distribution of the known sequence divergence rate. (c) Proportion and distribution of repeating elements.

This suggests that signal transduction-related genes play an important role in *B. multicinctus* (Figure 2). In addition, *B. multicinctus* genes were enriched in twelve metabolic pathways. The most enriched one was Lipid metabolism, and the least enriched one was Biosynthesis of other secondary metabolites.

## Data validation and quality control

We conducted a BUSCO (v5.2.2) (RRID:SCR_015008) assessment on the assembly to evaluate its integrity [7]. The assembly captured 90.9% of complete BUSCOs in the vertebrata_odb10 dataset (Figure 3).

To construct a phylogenetic tree, we screened closely related species, including *Anolis carolinensis*, *Chelonia mydas*, *Danio rerio*, *Deinagkistrodon acutus*, *Gallus gallus*, *Homo sapiens*, *Mus musculus*, *Ophiophagus hannah*, *Python bivittatus*, *Xenopus tropicalis*, and *Alligator mississippiensis*. Our data is consistent with previous studies and can be used to construct a phylogenetic tree that clusters closely related species (Figure 4) [8].

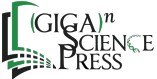



**Figure 2.** Gene annotation information of *B. multicinctus*. (a) KEGG enrichment of *B. multicinctus*. (b) Gene Ontology (GO) enrichment of *B. multicinctus*.

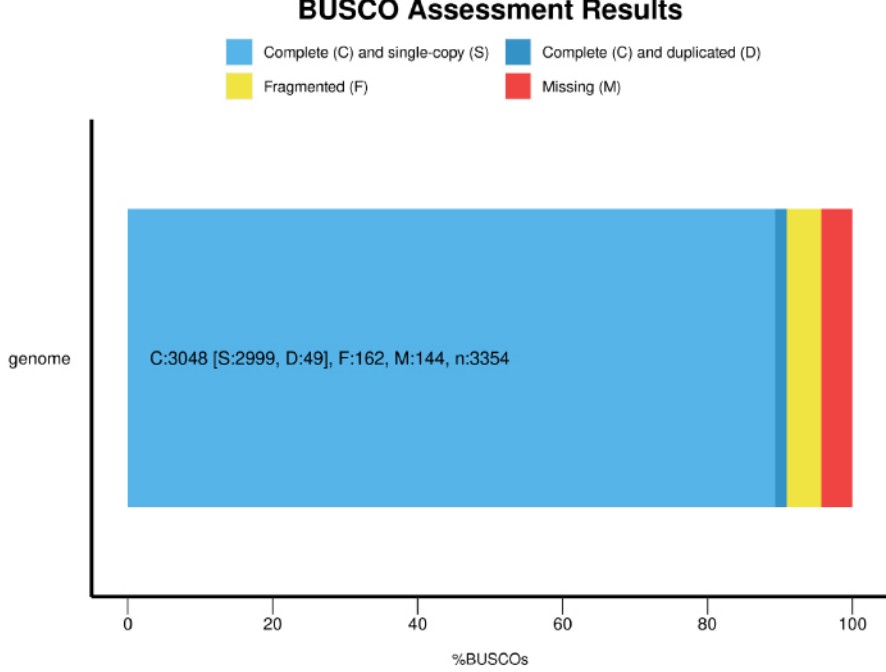

**Figure 3.** BUSCO Assessment result of our *B. multicinctus* genome.

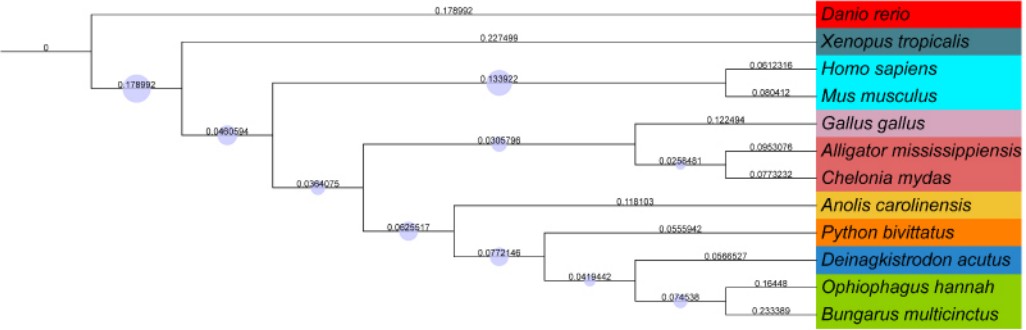

**Figure 4.** Phylogenetic tree reconstructed using nuclear genome single-copy genes.

## Methods

Detailed stepwise protocols are gathered in a protocols.io collection (Figure 5) [9] and summarised here.

### Sample collection and sequencing

*B. multicinctus* specimens were collected from Beiliu Longgukeng, Guangxi, and immediately transferred to dry ice for quick freezing. The samples were then stored at −80 °C. High-molecular-weight DNA was isolated using the protocol described by Wang *et al.* [10], and an stLFR co-barcoding DNA library was constructed using the MGIEasy stLFR Library Prep Kit (MGI, China). The libraries were sequenced using a BGISEQ-500 sequencer

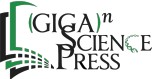

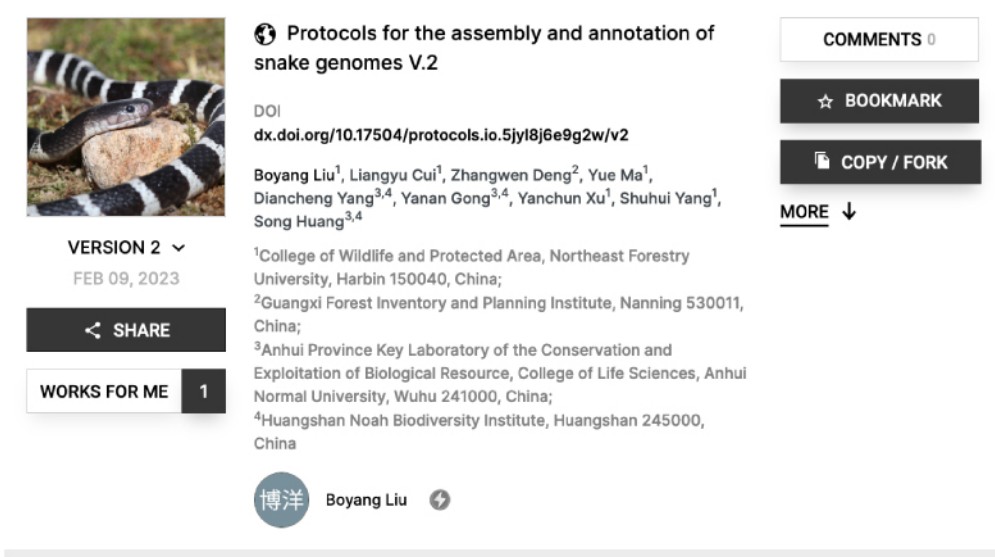

**Figure 5.** Protocols.io protocol collection for snake genome assembly [9]. https://www.protocols.io/widgets/doi?uri=dx.doi.org/10.17504/protocols.io.5jyl8j6e9g2w/v2

(RRID:SCR_017979) [11]. In addition, genomic DNA was isolated using the AxyPrep genomic DNA kit (AxyPrep, USA) for whole-genome sequencing.

We extracted the total RNA using the TRlzol reagent (Invitrogen, USA) following the manufacturer's protocol. RNA quality, purity, and quantity were assessed using the Qubit 3.0 fluorometer (Life Technologies, USA) and the Agilent 2100 Bioanalyzer System (Agilent, USA). The cDNA libraries were generated by reverse-transcribing RNA fragments of 200–400 bp. All experimental procedures were approved by the Institutional Animal Care and Use Committee of Northeast Forestry University.

### Genome assembly, annotation and assessment

The stLFR sequencing data obtained from the many-banded krait were subjected to assembly using Supernova (v2.1.1, RRID:SCR_016756) [12]. To improve the quality of the assembly, GapCloser (v1.12-r6, RRID:SCR_015026) and redundans (v0.14a) [9] were utilized for gap filling and redundancy removal, respectively, by incorporating the whole genome sequencing data.

To identify known repeat elements in the genome of the many-banded krait, Tandem Repeats Finder [13], LTR_FINDER (RRID:SCR_015247) [11], and RepeatModeler (v2.0.1, RRID:SCR_015027) [14] were utilized. RepeatMasker (v3.3.0, RRID:SCR_012954) [15] and RepeatProteinMask v3.3.0 [16] were employed for repeat element annotation. Protein-coding genes were predicted using *de novo*, homology-based, and transcript-mapping approaches. The *de novo* gene prediction was performed using Augustus (v3.0.3, RRID:SCR_008417) [17]. RNA-seq data were filtered using Trimmomatic (v0.30, RRID:SCR_011848) [18], and transcripts were assembled based on clean RNA-seq data using Trinity (v2.13.2, RRID:SCR_013048) [19] for RNA-seq-based prediction. PASA v2.0.2 [20] was utilized to align transcripts against the many-banded krait genome to obtain gene structures. Our homology-based prediction was performed by mapping protein sequences

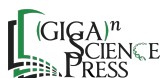

of the UniProt database (release-2020_05) of *Pseudonaja textilis*, *Crotalus tigris*, *Thamnophis elegans*, and *Notechis scutatus* to the *B. multicinctus* genome using Blastall v2.2.26 [21]. Gene models were predicted by analyzing the alignment results using GeneWise (v2.4.1, RRID:SCR_015054) [22]. Finally, the MAKER pipeline (v3.01.03, RRID:SCR_005309) [23] was employed to generate the final gene set, which represented RNA-seq, homology, and *de novo* predicted genes.

To perform functional annotations, a BLAST search (RRID:SCR_004870) was conducted against several databases, including SwissProt [24], TrEMBL [24], and KEGG [25], with an E-value cut-off of 1e-5. Furthermore, InterProScan (v5.52-86.0, RRID:SCR_005829) [26] was used to predict motifs, domains, and GO terms.

The genome completeness was evaluated by analyzing sets of BUSCO v5.2.2 using genome mode and lineage data from vertebrata_odb10 [27], following the standard scientific methodology.

To reconstruct the phylogenetic tree, OrthoFinder (v2.3.7, RRID: SCR_017118) [28] was used to search for single-copy orthologs among the protein sequences of *Anolis carolinensis* (GCA_000090745.2), *Chelonia mydas* (GCA_015237465.2), *Danio rerio* (GCA_000002035.4), *Deinagkistrodon acutus* [29], *Gallus gallus* (GCA_016699485.1), *Homo sapiens* (GCA_000001405.29), *Mus musculus* (GCA_000001635.9), *Ophiophagus hannah* (GCA_000516915.1), *Python bivittatus* (GCA_000186305.2), *Xenopus tropicalis* (GCA_000004195.4), and *Alligator mississippiensis* (GCA_000281125.4). The number of orthogroups of all species was 7,788.

## REUSE POTENTIAL

Venomous animals have fascinated and influenced humans since ancient times, and the venom gland is a special evolutionary mechanism that snakes have developed to adapt to their ecological environment [30]. In recent years, ecosystems have changed due to climate variations, and toxic species threaten not only humans but also native species and livestock [31, 32]. Therefore, it is crucial to collect genomic resources of venomous snakes and explore the formation mechanism of venom glands and venom production.

Genome assemblies of reptiles, including snakes, have always been challenging to generate. However, Xu *et al.* recently published an article on the origin of neurotoxins in the Elapidae family based on a high-quality genome assembly of the many-banded krait [29]. Using third-generation sequencing and Hi-C, Xu *et al.* assembled the many-banded krait genome to the chromosome level, achieving a BUSCO score of 94.6% and a scaffold N50 of 149.80 Mbp. Our assembly resulted in a BUSCO score of only 90.9%. Although our assembly did not achieve the same level of genome continuity as Xu *et al.*, we obtained a relatively complete genome of the many-banded krait using stLRF second-generation sequencing data. Being sampled from a different geographic location provides a genomic resource for future research exploring the evolution and origin of reptilian species, including snakes.

Our data can be combined with already published and new venomous snake genome data to reconstruct the evolutionary history of venomous snakes and other reptiles. Our genome data can also be used in venomics research to explore toxic gland genes and the mechanism of toxic gland production.



## DATA AVAILABILITY

The data that support the findings of this study have been deposited into CNGB Sequence Archive (CNSA) [33] of China National GeneBank DataBase (CNGBdb) [34] with accession number CNP0004003. The data are also hosted in NCBI with accession number PRJNA934116. Additional data is available in the GigaDB repository [35].

## LIST OF ABBREVIATIONS

GO, Gene Ontology; KEGG, Kyoto Encyclopedia of Genes and Genomes; LINEs: Long Interspersed Nuclear Elements; LTRs: Long Terminal Repeats; SINEs: Short Interspersed Nuclear Elements; TE, transposable element.

## DECLARATIONS

### Consent for publication

All experimental procedures were approved by the Institutional Animal Care and Use Committee of Northeast Forestry University (2023048).

### Competing Interests

The authors declare no conflict of financial interests.

### Authors' contribution

SH, SY, and YX designed and initiated the project. ZD, DY, and YG collected the samples. BL, LC, and YM performed the DNA extraction and data analysis. BL, LC, ZD and TL wrote the manuscript. All authors read and approved the final manuscript.

### Funding

This work was supported by the Fundamental Research Funds for the Central Universities (No. 2572020DY02) and the Guangdong Provincial Key Laboratory of Genome Read and Write (grant No. 2017B030301011). This work was also supported by China National GeneBank (CNGB).

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
