## [Editor Report]

Comments to the AuthorThe many-banded krait, Bungarus multicinctus is a highly venomous snake distributed across South and Eastern Asia. To help better understand the evolution of B. multicinctus, and also provides a molecular basis for the understanding venom production a 1.51Gb in size reference genome was sequenced and described here. This data can be combined with already published and new venomous snake genome data to construct the evolutionary history of venomous snakes and other reptiles. After submission another group published a genome of a different B. multicinctus individual, but after clarification on some parts of the genome assembly process as well as mentioning the other genome, it is useful to share this data to the community.

---

## [Reviewer Report]

Comments on revised manuscriptThe revised manuscript should not be satisfied with. For Q3, if authors can not decide which result to choose, they shoudl use repeatmasker to integrate all annotation. For Q4 and Q6, answers are feckless. If the author want to provide a certified data for sharing, they should ensure the quality. Authors enhanced the importance of their data for venom analysis, they should prove their value themselves. And as some chromosome-level high quailty krait genomes have been released, they should compare them at least in the phylogenetic analysis.

---

## [Reviewer Report]

Reviewer name and names of any other individual's who aided in reviewer Xu JiangDo you understand and agree to our policy of having open and named reviews, and having your review included with the published papers. (If no, please inform the editor that you cannot review this manuscript.)YesIs the language of sufficient quality?YesPlease add additional comments on language quality to clarify if needed
Are all data available and do they match the descriptions in the paper? YesAdditional CommentsAre the data and metadata consistent with relevant minimum information or reporting standards? See GigaDB checklists for examples <a href="http://gigadb.org/site/guide" target="_blank">http://gigadb.org/site/guide</a>YesAdditional CommentsIs the data acquisition clear, complete and methodologically sound?YesAdditional CommentsIs there sufficient detail in the methods and data-processing steps to allow reproduction?YesAdditional CommentsIs there sufficient data validation and statistical analyses of data quality? NoAdditional CommentsIs the validation suitable for this type of data?NoAdditional CommentsIs there sufficient information for others to reuse this dataset or integrate it with other data?YesAdditional CommentsAny Additional Overall Comments to the AuthorIn this work, Boyang Liu et al used a stLFR method for the assembly of a Bungarus multicinctus genome. This work is technique sound for the innovation application fo stLFR which is cost low and sequencing fast. Meanwhile, the B.multicinctus is a very important species in snake as its highly lethal venom. Before further consideration, I think there are some issues should be answered. 1. Suprenova was used for this work, as I know, the supernova can export the phased info of the assembly. And I also noticed that authors used the redundans (v0.14a) for the removal of reduancy. So I am very intereset at how many redundancy was removed, and I didn't find this part of information in the work.  2. For Table 1. I noticed that the total length of contig and scaffold are same(both are 1621955402) , but the GC content of these two assemblies are highly different (39.6% vs 37.8%). May authors give some more explanation or discussion? 3. For Table 2 and Table 3, the statistics were not consistency, please choose one. 4. In the abstract, authors declaimed that "and also provides a molecular basis for the genes of the venom glands." However, in the Fig. 2, i don't find the toxicity venom information. As my knowledge, these families/pathway were enriched in B.multicinctus genome. I think the authors can check their annotation and RNA-seq data.  5. Fig 4 is not suitable. The species selection is a little sloppy.  6. There is a high quality genome reference of B.multicinctus released and deposited at http://www.gpgenome.com/species/148. However, this snake is not from guangxi. And I think the author can compare it with this work, the paper can be fetched from https://doi.org/10.1016/j.apsb.2022.11.015. How to use the database, please refer to https://doi.org/10.1007/s11427-021-1968-7.RecommendationMajor Revision

---

## [Reviewer Report]

Reviewer name and names of any other individual's who aided in reviewer Nazila GodfreyDo you understand and agree to our policy of having open and named reviews, and having your review included with the published papers. (If no, please inform the editor that you cannot review this manuscript.)YesIs the language of sufficient quality?NoPlease add additional comments on language quality to clarify if needed
The writing needs to be improved significantly.Are all data available and do they match the descriptions in the paper? NoAdditional CommentsThere is only one Biosample submitted for this project while in methods is mentioned samples. It is not clear if the biosample is for DNA or RNA. Following sequencing data depends on this too. Are the data and metadata consistent with relevant minimum information or reporting standards? See GigaDB checklists for examples <a href="http://gigadb.org/site/guide" target="_blank">http://gigadb.org/site/guide</a>NoAdditional CommentsFasta files required for genome assembly and BUSCO files are missing.Is the data acquisition clear, complete and methodologically sound?NoAdditional CommentsThe genome assemblies that are used need to be properly cited.Is there sufficient detail in the methods and data-processing steps to allow reproduction?NoAdditional CommentsIs there sufficient data validation and statistical analyses of data quality? NoAdditional CommentsIs the validation suitable for this type of data?YesAdditional CommentsIs there sufficient information for others to reuse this dataset or integrate it with other data?NoAdditional CommentsAny Additional Overall Comments to the AuthorIntroduction needs to be elaborated heavily. The importance of study needs to be clarified. Why only %6 of the bite cases is treated with antivenom? Is that because bites usually happen in remote areas, or we don’t have antivenom? It is confusing. Why genome sequencing is necessary for designing an antivenom?  Main content section needs to be expanded and elaborated. It is not clear why at the end of this section they made such a statement: “This suggests that the significant expansion of repeating elements is an important manifestation of species differences”. Explain in the text.
RecommendationMajor Revision